# The Role of Sphingolipid Signaling in Oxidative Lung Injury and Pathogenesis of Bronchopulmonary Dysplasia

**DOI:** 10.3390/ijms23031254

**Published:** 2022-01-23

**Authors:** Jaya M. Thomas, Tara Sudhadevi, Prathima Basa, Alison W. Ha, Viswanathan Natarajan, Anantha Harijith

**Affiliations:** 1Department of Pediatrics, Case Western Reserve University, Cleveland, OH 44106, USA; jmt230@case.edu (J.M.T.); txs671@case.edu (T.S.); pxb421@case.edu (P.B.); axh780@case.edu (A.W.H.); 2Department of Biochemistry and Molecular Genetics, University of Illinois at Chicago, Chicago, IL 60607, USA; 3Department of Pharmacology and Regenerative Medicine, University of Illinois at Chicago, Chicago, IL 60607, USA; visnatar@uic.edu; 4Department of Medicine, University of Illinois at Chicago, Chicago, IL 60607, USA

**Keywords:** bronchopulmonary dysplasia, ceramide, S1P, ROS, sphingolipid signaling, mitochondrial dysfunction

## Abstract

Premature infants are born with developing lungs burdened by surfactant deficiency and a dearth of antioxidant defense systems. Survival rate of such infants has significantly improved due to advances in care involving mechanical ventilation and oxygen supplementation. However, a significant subset of such survivors develops the chronic lung disease, Bronchopulmonary dysplasia (BPD), characterized by enlarged, simplified alveoli and deformed airways. Among a host of factors contributing to the pathogenesis is oxidative damage induced by exposure of the developing lungs to hyperoxia. Recent data indicate that hyperoxia induces aberrant sphingolipid signaling, leading to mitochondrial dysfunction and abnormal reactive oxygen species (ROS) formation (ROS). The role of sphingolipids such as ceramides and sphingosine 1-phosphate (S1P), in the development of BPD emerged in the last decade. Both ceramide and S1P are elevated in tracheal aspirates of premature infants of <32 weeks gestational age developing BPD. This was faithfully reflected in the murine models of hyperoxia and BPD, where there is an increased expression of sphingolipid metabolites both in lung tissue and bronchoalveolar lavage. Treatment of neonatal pups with a sphingosine kinase1 specific inhibitor, PF543, resulted in protection against BPD as neonates, accompanied by improved lung function and reduced airway remodeling as adults. This was accompanied by reduced mitochondrial ROS formation. S1P receptor1 induced by hyperoxia also aggravates BPD, revealing another potential druggable target in this pathway for BPD. In this review we aim to provide a detailed description on the role played by sphingolipid signaling in hyperoxia induced lung injury and BPD.

## 1. Introduction

Bronchopulmonary dysplasia (BPD) was initially characterized by Northway and colleagues in 1967 [1]. Since the first description, it was a challenge to define and reconcile various definitions of the same in the literature [2]. A recent definition of BPD by Shennan et al. defines the need for supplemental oxygen at 36 weeks of postmenstrual age. This correlates well with the long-term respiratory morbidity seen at two years of age [3]. Apart from premature infants who need prolonged oxygen supplementation, BPD also occurs in premature infants who require short-term oxygen supplementation [4]. The underdeveloped antioxidant defense system [5], along with surfactant deficiency in a developing lung exposed to various injurious agents [6], make BPD a disease with a complex pathogenesis. BPD is characterized by the presence of respiratory signs along with histological observations of alveolar simplification, reduced pulmonary vascularization, parenchymal fibrosis, and edema [7,8]. In United States, 10–40% of premature infants born weighing between 500 and 1500 g are diagnosed with BPD in their life, leading to approximately 10,000 infants being affected per year [9]. BPD can be considered as a disease with multifactorial etiology [10]. Diverse signaling molecules, such as VEGF, miR-219, CD44, etc., [11,12] were associated with the pathogenesis of BPD. Furthermore, there is a strong role for genetics in the pathogenesis of BPD as well [13]. Studies on identical and nonidentical twins affirmed this observation [13], in addition to the findings that BPD is more severe in males compared to females [14,15]. A high degree of congruence in the severity of BPD was noted among identical twins compared to nonidentical twins [13]. Single nucleotide polymorphisms (SNPs) in genes such as angiotensin-converting enzyme [16], and *SPOCK2*, a marker downstream of surfactant protein D gene [17], are associated with BPD. Inflammatory conditions both in the pre- and postnatal period such as chorioamnionitis and various forms of neonatal sepsis are linked to BPD [18,19,20,21]. All these findings suggest that BPD is a heterogeneous disease where different genes and molecular signaling pathways converge. Therapy for BPD include use of anti-inflammatory agents, steroid therapy, less intense ventilator strategies, vitamin A, diuretics, and caffeine [22,23,24,25,26]. Despite all these therapeutical strategies, a cure for this disease is not in sight, with no FDA approved drugs(s) available to treat the condition.

In this context, recent studies suggest a key role for sphingolipids in the pathogenesis of BPD [27,28,29,30,31]. For a prolonged period, sphingolipids were solely considered as the structural constituents of cell membrane [32,33]. Lately, its role as regulatory molecules in various functions such as immune response, cell proliferation, differentiation, apoptosis, and angiogenesis were identified [34]. Dysregulation in sphingolipid signaling pathway is noted in both pulmonary and nonpulmonary diseases [35,36,37,38,39,40]. Interestingly, alteration in sphingolipid signaling axis, as manifested by increased levels of ceramides S1P and S1P synthesizing enzyme, sphingosine kinase 1 (SPHK1), was observed in lung and tracheal aspirates of BPD patients [29,30,41].

Oxygen therapy is inevitable for the survival of premature infants [42]. However, long-term therapy with oxygen or exposure to hyperoxia of underdeveloped lungs with a poor antioxidant defense [5] can generate reactive oxygen species (ROS), which contributes to the pathogenesis of BPD [43]. BPD is a multifactorial disease leading to inflammation, cellular apoptosis and extracellular matrix remodeling affecting lung growth, function, immunity, alveolarization, and vascularization affecting repair and regeneration (Figure 1). The role of sphingolipid signaling in the generation of ROS and apoptosis was demonstrated [44]. Recent studies highlighted a potential link between sphingolipid signaling inducing ROS formation and BPD [29]. Pathological role of alveolar epithelial type II cells came under focus recently [45,46]. In our recent study using human primary small airway epithelial cells, we identified that inactivation or inhibition of SPHK1 suppressed hyperoxia (HO)-induced mitochondrial ROS production [29]. Recent studies point to a link between sphingolipids and mitochondrial dysfunction. In this review, we provide a comprehensive review of the literature giving a better insight into sphingolipid metabolism, mitochondrial dysfunction, and the role of sphingolipid-signaling-induced ROS leading to BPD.

## 2. Sphingosine-1-phosphate Synthesis, Metabolism, and Signaling

Sphingolipids are a class of naturally occurring lipids with a sphingosine or dihydrosphingosine backbone that are mainly located in the plasma membrane in association with lipid rafts [47,48]. Various sphingolipid-derived mediators play a key role in both physiological and pathological processes [49,50,51,52]. Sphingolipid de novo synthesis (Figure 2) starts at the endoplasmic reticulum (ER) with the formation of the long chain base (LCB), which is the characteristic backbone of all SLs. Sphingomyelin, the major sphingolipid, is synthesized by the condensation of serine and palmitoyl coenzyme A. This reaction is catalyzed by the enzyme serine palmitoyl transferase (SPT) through a rate limiting step, leading to the formation of 3–keto-dihydro sphingosine [53,54]. There are three subunits for SPT—SPTLC1, SPTLC2, and SPTLC3—and each subunit has a specific action. A recent finding is that the SPTLC3 subunit could produce a methyl-branched sphingoid base [55]. The 3–keto-dihydro sphingosine is reduced to dihydro sphingosine and acylated to dihydroceramide by ceramide synthase, and subsequently, it is desaturated to ceramide. Ceramide synthase includes a group of six enzymes. Ceramide synthase and elongase are the major enzymes involved in long chain fatty acid ceramide synthesis. Recent studies identified that diacylglycerol O-acyltransferase 1 and diacylglycerol O-acyltransferase 2 could acetylate ceramide at 1-OH position [32,56,57]. Ceramide is converted to sphingomyelin or hydrolyzed to sphingosine through ceramidases or channeled to glycosphingolipids [32]. Sphingosine, the primary base present in all sphingolipids, is generated from ceramide by the enzyme ceramidase [32]. Sphingosine in mammalian cells is converted to sphingosine-1 phosphate (S1P) [58].

S1P is an important bioactive signaling lipid mediator that is generated by phosphorylation of ceramide derived sphingosine by the enzyme sphingosine kinase. There are two isoforms for this enzyme: sphingosine kinase 1 (SPHK1) and sphingosine kinase 2 (SPHK2) [34,50]. SPHK1 is predominantly located in the cytosol, and it is activated by phosphorylation at serine 225 [59,60]. Once activated, SPHK1 is translocated from the cytosol to the plasma membrane and catalyzes the formation of S1P. SPHK1 can be activated by growth factors, hormones, lipopolysaccharides hypoxia, and hyperoxia [29,30,60]. In contrast to SPHK1, SPHK2 is located in intracellular membranes in addition to the nucleus [59]. Nuclear SPHK2 regulates gene expression by inhibiting histone deacetylase through nuclear S1P generation [61]. Mitochondrial SPHK2 helps to maintain the cytochrome oxidase complex [62]. SPHK1 was recently noted to play a role in the formation of mitochondrial ROS, and new data are emerging on the role of sphingolipid and mitochondria in promoting lung injury [29].

S1P is rapidly degraded by the enzymes S1P lyase (S1PL), S1P phosphatase 1 and 2, and lipid phosphate phosphatases [34]. Intracellularly generated S1P in endothelial cells and other mammalian cells is also exported out of the cell by specific transporters, such as spinster 2 (SPNS2) and ABCA3 [63]. Intracellular concentration of S1P is critically maintained by the action of synthesizing and degrading enzymes. The released S1P and S1P bound to HDL in plasma act as ligands for five G-protein coupled receptors, S1P receptor 1 (S1PR1)-5. S1P receptors are differentially expressed in different cell types [34]. Thus, S1P signaling can activate different signal transduction pathways in each cell type, resulting in varied outcomes [51,59]. Additionally, S1P generated inside the cells also signals independently of S1PR1-5 by binding to and modulating various targets, such as HDAC1/2 and hTERT [61].

## 3. Sphingolipids in Diseases

Various cellular and physiological functions, such as proliferation, apoptosis, differentiation, cytoskeletal organization, adhesion, tight junction assembly, migration, and morphogenesis are modulated by ceramides and SPHK1s/S1P/S1PRs axis [34,57]. Thus, an aberrant regulation in S1P signaling can result in pathological effects.

Role of sphingolipids in the pathogenies of lung disease is reviewed here, stressing its role in neonatal lung diseases. S1P has a crucial role in regulating pulmonary endothelial barrier integrity both in physiological and pathological conditions [64]. Pulmonary edema is the reason for morbidity and mortality in acute lung injury (ALI) and acute respiratory distress syndrome [65]. In ALI, there is increased activity of acid sphingomyelinase (ASMase), and studies showed that ASMase knock out mice improved pulmonary vascular integrity [66,67]. S1P through S1PR1 stabilizes the endothelial cell barrier [64]; however, activation of SP1R2 and S1PR3 results in an imbalance of endothelial cell barrier [64,68]. Thus, depending on receptor subtypes, S1P can cause either physiological or pathological effects in barrier regulation and integrity.

S1P is involved in the cascade of pathogenic events leading to asthma [69]. Airway hyperresponsiveness (AHR) and cholinergic airway contraction are the key features in allergic asthma [70]. S1P, by stimulating S1PR2, can aggravate AHR and promote airway remodeling [71]. S1P also induces chemotaxis of inflammatory cells and worsens the severity of asthma [72]. Role of sphingolipids, especially those of ceramides, in regulating cystic fibrosis (CF) was reported. Ceramides regulate development of infections and inflammations associated with CF [73,74]. Concentration of ceramides was increased in the lower airway epithelium of lungs of patients with CF [75]. In animal models of CF, a reduction in ceramide levels was associated with reduced severity of lung infections [76,77,78]. In preclinical models of pulmonary fibrosis and pulmonary artery hypertension (PAH), genetic deletion of *Sphk1* or inhibition of SPHK1 (but not SPHK2) activity with PF543, a small molecule inhibitor of SPHK1, reduced bleomycin-induced development of lung fibrosis or hypoxia-induced PAH in mice. These studies clearly showed that targeting SPHK1 is beneficial in ameliorating experimental lung fibrosis or PAH in murine models [79,80,81].

S1P was found to have a pathological role in organs beyond lungs. S1P is stored in erythrocytes and plays a role in the sickling of these cells in sickle cell disease [82]. Increased SPHK1 activity in erythrocytes along with increased S1P in blood of sickle cell disease patients with SCD was demonstrated. Sickling was noted to be independent of S1P receptor activation in isolated erythrocytes, as shown by ex vivo experiments. It was shown in murine models that S1P was greatly increased in cardiac tissue postmyocardial infarction. This was associated with a significant increase in cardiac sphingosine kinase-1 (SPHK1) and S1P receptor 1 (S1PR1) expression. FTY720 is a nonspecific S1PR antagonist that also blocks SPHK1 and ceramidase, reduces cardiac inflammation, and improves cardiac remodeling and function postmyocardial infarction [83].

Nonlung diseases such as cancer, atherosclerosis, diabetes, and multiple sclerosis are also associated with altered sphingolipid signaling [84,85,86,87]. There is an increased expression of S1P receptors and enzyme SPHK1 in cancerous tissues [84,88]. S1P could promote neovascularization, epithelial mesenchymal transition, and invasiveness in tumors [89,90]. S1P also causes both pro- and antiatherogenic effect. Atherogenic effect is mediated through S1PR3, and antiatherogenic effect is mediated through S1PR1 [85]. In patients with multiple sclerosis there is increased level of S1P in the cerebrospinal fluid and brain parenchyma. Fingolimod (FTY720), an S1P receptor modulator, is an FDA-approved drug for the treatment of multiple sclerosis [87]. FTY720 remains a promising drug in the treatment of Type II diabetes mellitus [91]. In a mouse model of Type II diabetes mellitus, S1P upregulates the number of pancreatic β cells by increasing its proliferation and inhibiting apoptosis [92]. No clinical trials are reported. Small molecule inhibitors involved in S1P signaling axis are included in Table 1.

Recently the role of deoxysphingolipids came to light in various pathological disorders [105]. As explained earlier, SPT, physiologically conjugates l-serine and palmitoyl-CoA to form 3-keto-phinganine, which is converted to sphinganine [54,106]. However, in an alternative reaction, SPT promotes binding of L-alanine with palmitoyl-CoA to form 1-deoxysphinganine or conjugation of glycine with palmitoyl-CoA to form 1-deoxymethylsphinganine [107,108]. Deoxyshingolipids lacks C1-OH-group present in canonical sphingolipids. Due to the absence of C1-OH-group, deoxysphingolipids are not terminally degraded by S1P-lyase, leading to their accumulation in the tissues [106,108]. L-serine depletion leads to formation of 1-deoxysphingolipids, which promotes senescence [109,110]. The toxic effects of deoxysphingolipids causes mitochondrial toxicity and are suggested to cause hereditary sensory autonomic neuropathy type 1 [107]. The role of deoxysphingolipids in diabetic neuropathy is under investigation [111]. Over expression of alpha-synuclein in Parkinson’s disease can cause accumulation of deoxysphingolipids and promotes neurotoxicity [55,112]. A recent study identified that cytochrome P4504F enzymes could metabolize 1-deoxyshingolipids and are a promising therapeutic target in Type II diabetes, hereditary sensory autonomic neuropathy type 1, and diabetic sensory neuropathy [113].

## 4. Ceramide, SphK1/S1P/S1PR1 Signaling Axis in BPD Pathogenesis

BPD is a devastating disease that affects more than half of preterm infants born at a gestational age of <26 weeks [4]. Hence, it is critical to delineate the molecular mechanisms in the pathogenesis of BPD to identify new therapeutic targets. Recent research in the field of BPD discovered many mediators associated in its pathogenesis [10]. Significance of sphingolipid metabolites in lung diseases such as asthma, chronic obstructive pulmonary disease, and cystic fibrosis emphasizes the importance of sphingolipids in neonatal lung injury [69,75,114]. Recent studies identified ceramides and S1P as the key players of sphingolipid biology in the pathogenesis of BPD [27,29,30,41].

Ceramides induce proinflammatory cytokine synthesis, promote apoptosis, and cause cell cycle arrest. These effects of ceramides are manifested in BPD as increased apoptosis of pulmonary cells. Animal studies identified increased concentration of ceramide in bronchoalveolar lavage (BAL) during the initial days of hyperoxia exposure. In murine model of hyperoxia normalization of ceramide concentration by supplementation with D-sphingosine-ameliorated hyperoxia-induced arrest of alveolar development [40]. Findings in animal studies were in corroboration with the findings in humans. Ceramide levels were increased in tracheal aspirates (TAs) of preterm infants immediately after one day of oxygen supplementation and mechanical ventilation. Also, there was significant difference in ceramide concentration in TAs of preterm infants who developed BPD when compared to that of normal infants [41].

Consistent with this finding of ceramide in BPD, an increase in S1P concentration is also associated with BPD pathogenesis. An increased level of S1P in TAs of neonatal infants with BPD was observed, which was substantiated in an animal model of hyperoxia [30]. In both in vivo and in vitro models of hyperoxia, attenuating SPHK1/S1P/S1PR1 signaling axis improves hyperoxia-induced lung injury, manifested by decreased inflammatory cytokine levels, maintenance of alveolar integrity, and improved alveolarization [27,29,30,31]. Targeting SPHK1/S1P/S1PR1 axis also contributed to extended protection in long-term sequela of airway remodeling by improving lung development and function.

Both SPHK1 and SPHK2 isoenzymes phosphorylate sphingosine to generate S1P [34,115]. However, it is only SPHK1, and not SPHK2, that is upregulated in hyperoxia. Subsequently, only the *Sphk1* knocked out mice exhibited reduced hyperoxia-induced lung injury compared to that of wild type mice [27]. Moreover, the SPHK1 activity inhibitor, PF543, ameliorated hyperoxia-induced lung injury, accompanied by restoration of E-cadherin expression in airway epithelium that was reduced following hyperoxia [29].

BPD is characterized by rigid lung parenchyma and an increased respiratory resistance that reduces lung capacity [8]. Preterm infants suffer from poor lung compliance due to increased lung parenchymal rigidity [116]. Increased cross-linking of the extracellular matrix protein collagen by lysyl oxidase (LOX) contributes to lung parenchymal rigidity [117]. There was increased expression of lysyl oxidase (LOX), in tracheal aspirates of preterm infants with BPD. The finding that SPHK1/S1P signaling axis is involved in transcriptional activation of LOX endorses the link between SPHK1/S1P/LOX signaling axis in the pathogenesis of BPD [30].

In our recent study, we identified S1PR1 over-expression in lungs of wild type mice exposed to hyperoxia (75%). We demonstrated that partial deletion of S1PR1 offers protection from BPD in the murine model [31]. Aberrant lung vasculature is one of the hallmarks of BPD [118], and partial deletion of S1PR1 promoted angiogenesis with improved pulmonary vasculature [31]. Increased expression of S1PR1 under hyperoxia inhibited retinal angiogenic sprouting [119]. Based on these results we hypothesize that S1PR1 modulator, FTY720, could be a promising drug in the treatment of BPD. FTY720 is yet to be proven to be effective in the treatment of BPD in animal models. Though approved by FDA for use in children, FTY720 is associated with cardiovascular side effects, and hence, the safety profile in preterm neonates must be established prior to extensive clinical studies.

To date, the various findings in BPD in relation to sphingolipids can be summarized as follows: (i) preterm babies with increased ceramide and S1P levels in TAs are more likely to develop BPD; (ii) SPHK1 enzyme expression is increased both in patients with BPD and the murine hyperoxia model of BPD, and the *Sphk1* knockout model offers protection; (iii) SPHK1 enzyme inhibitor, PF543, provides protection against hyperoxia-induced lung injury; and (iv) S1PR1 is upregulated in a mouse model of hyperoxia and promotes pathogenesis of BPD [27,29,30,31,41]. These findings substantiate that SPHK1/S1P/S1PR1 axis is a druggable target for the treatment of BPD.

## 5. Reactive Oxygen Species and Antioxidant Defense Mechanisms

ROS include oxygen radicals, such as superoxide (O2 −), peroxyl (RO2•), hydroxyl (•OH−), alkoxyl (RO.), hypochlorite (OCl−), and nonradicals, such as hypochlorous acid (HOCl), ozone (O3), singlet oxygen (O2), and hydrogen peroxide (H2O2). Among these, superoxide anion (O2 −) is the central molecule involved in the production of majority of other ROS [120]. Physiologically, under basal condition, there is a continuous production of ROS through various biological processes. ROS production can also be stimulated by various external stimuli [121]. While mitochondrial electron transport is the main source of mitochondrial ROS production, other major cellular sources of ROS production are NADPH oxidase (NOXs) and Dual Oxidase (DUOXs) [122].

Intracellular ROS generation and signaling have both physiological and pathological functions and were implicated in cellular activities such as phagocytosis, cell signaling, cell growth, differentiation, apoptosis, and necrosis. By regulating all these cellular processes, ROS help to maintain tissue homeostasis [123]. In embryo, ROS are essential for normal development [121]. By contrast, excess production of ROS can cause imbalance in redox homeostasis, ultimately leading to oxidation of biological molecules, such as proteins, lipids, carbohydrates, and nucleic acid, disrupting their structure and cellular functions [124,125,126,127]. Oxidation of nucleic acids can cause DNA fragmentation and eventually lead to either necrosis or apoptosis [124]. Both amino acids and protein backbone are oxidized by ROS, resulting in various post-translational modifications. Sulfur containing amino acids such as cysteine and methionine are more prone to oxidation induced by ROS [126,127]. Polyunsaturated fatty acids present in cell membrane lipids are highly susceptible to the attack of ROS, resulting in lipid peroxidation and activation of extrinsic and intrinsic pathways of apoptosis. Lipid peroxidation also generates reactive aldehydes, such as hydroxynonenal phospholipids with short chain aldehydes that are very reactive and form Schiff’s base and Michael adducts with proteins, DNA, and phosphatidylethanolamine in biological systems. Modulation of macromolecules by ROS and lipid-derived fatty aldehydes affects the normal cellular function and ultimately leads to adverse pathological outcomes [128].

ROS are produced in two scenarios: (i) as the byproduct of cellular reactions such as oxidative phosphorylation in mitochondria and in peroxisomes; (ii) as a direct product of specific cellular functions such as phagocytosis and signal transduction [129]. NOX’s protein system in phagocytosis is the first identified enzyme mechanism that is mainly involved in the ‘intentional’ production of ROS. During phagocytosis, phagocytotic NOX2 is activated, leading to the production of hypochlorous acid (HOCl), an antimicrobial agent [130]. In the process of phagocytosis, some ROS may leak out of the phagosome to the cytoplasm and induce oxidative stress to the cell. Nonphagocytotic NOX proteins are involved in the production of ROS and signal transduction. Various growth factors and cytokines activate nonphagocytotic NOX system [131]. In addition to NOX family, cyclooxygenase, lipoxygenase, mitochondrial electron transport chain, cytochrome P450, and xanthine oxidase are also involved in ROS production in various metabolic processes [132]. NOX proteins are a multicomponent protein complex that includes NOX1, NOX2, NOX3, NOX4, and NOX5, as well as DUOX proteins, namely, DUOX1 and DUOX2. NOX2 is the prototype among NOX proteins and is widely distributed throughout the tissues. All NOX proteins are transmembrane proteins and have a conserved structure with NADPH- (or NADH) and FAD-binding domains in their C terminal [131]. They transfer electrons across membrane and reduce oxygen to superoxide. NOX proteins induce the reduction in molecular oxygen through NADPH and produce superoxide free radicals. Among the NOX proteins, NOX4 and DUOX1 and 2 mainly generate hydrogen peroxide and not superoxide [133].

Cells need to maintain concentration of ROS for the normal functioning of physiological processes, as the excess accumulation of ROS is detrimental. Maintenance of intracellular ROS at low levels is achieved by antioxidant systems; however, breakdown or overwhelming of the antioxidants results in a shift in the balance towards excess ROS accumulation in the cell [134]. Antioxidant enzymes have a pivotal role in maintaining redox homeostasis as it decomposes ROS. Superoxide dismutase, glutathione peroxidase, thioredoxin reductase, catalase, glutathione reductase, manganese superoxide dismutase and copper-zinc superoxide dismutase are the various enzymes involved in antioxidant defense [134,135]. Newborn infants are largely affected by the adverse effects of ROS due to poor development of enzymatic antioxidant defense mechanism [5]. Vitamin A, tocopherol, glutathione, coenzyme Q10, ascorbic acid, flavonoids, zinc, and selenium are components of the cellular nonenzymatic defense mechanisms [134,135].

## 6. Oxidative Stress and Mitochondrial Injury

BPD is an oxygen radical disease in neonatology, along with retinopathy of prematurity (ROP) and necrotizing enterocolitis [136]. In preterm infants, mechanical ventilation along with use of high oxygen concentration leads to increased ROS production that results in injury to multiple organs, especially lung parenchyma [43]. NOX proteins are predominantly involved in ROS production in hyperoxia exposure [137]. There is a recent report on association between C242T single nucleotide polymorphism in CYBA gene and BPD [137]. CYBA gene encodes the p22*^phox^* subunit of nicotinamide adenine dinucleotide phosphate NOX [138]. The role of NOX4 and NOX2 in hyperoxia-induced ROS production and migration of endothelial cells was described [139]. The impairment in postnatal alveolar type II cell development and increased alveolar septal elastin deposition, which is hyperoxia-induced, and NOX2 protein-mediated were diminished in NADPH oxidase *p47^phox^* null mice [140]. Hyperoxia-enhanced NOX4 expression in endothelial cells involves binding of NRF2 to antioxidant responsive elements in NOX4 [141]. Defective NRF2 can cause ROS-induced mitochondrial dysfunction and metabolic disorders [142]. The clinical use of classical antioxidants was limited due to various biochemical/physiological issues and requires additional studies [143]. As mentioned in the preceding section, in a healthy cell, oxidative stress induced by hyperoxia is balanced by antioxidant defense mechanisms, especially the enzymatic antioxidant defense system [134]. Compared to that of preterm newborns, full-term infants can resist oxidative stress [144]. The dearth of this defense mechanism in preterm infants make their lungs more vulnerable to the adverse effects of ROS [145]. ROS induces airway remodeling and promotes ER stress. ER stress subsequently leads to further apoptosis of alveolar type II (ATII) cells, induction of various inflammatory pathways, and impairment in angiogenesis [146].

ATII cells are the main source for the antioxidant enzyme system in lungs [147]. Hypoplasia of ATII cells in preterm infants make them defenseless to ROS-mediated injury [148]. ATII cells act as the stem cells that produce alveolar type I (ATI) cells and participate in the production of various bioactive substances, especially the surfactant [149]. ATII cells help to maintain alveolar capillary barrier integrity and alveolar function [147]. ROS induces apoptosis of ATII cells, and alveolar capillary integrity is lost, resulting in interstitial edema and gas exchange impairment [150].

Mitochondrial dysfunction was recently recognized to play a significant role in the pathogenesis of various diseases [29,151,152]. Its role in neonatal BPD is beginning to emerge, and the involvement of sphingolipids in propagating mitochondrial damage is yet to be elucidated in detail.

## 7. Mitochondrial Oxidative Stress in BPD

Mitochondria are the principal sites of ROS production in a cell [153]. In addition to generating ATP, mitochondria regulate numerous other activities, including cell proliferation, differentiation, transduction of calcium signaling, mitophagy, autophagy and apoptosis [154]. Recent evidence suggests that mitochondrial dysfunction caused by hyperoxia contributes to alveolar developmental arrest in BPD [155,156]. Although mitochondria of the neonatal lungs produce more ROS than cytoplasm and are more tolerant to hyperoxia [157], preterm babies subjected to higher oxygen exposure develop impaired mitochondrial function, affecting cell growth, differentiation, and tissue development, leading to BPD and central nervous system damage [158]. Mitochondria could be a potential target for specific intervention in lung disease, including BPD [159]. Use of mitochondria targeted antioxidants, such as MitoTEMPOL and mitoQ, can scavenge ROS [160] and prevent hyperoxia-induced lung injury by reducing NOX expression and/or activity [161], which needs further investigation.

Oxidative stress causes mitochondrial DNA (mtDNA) damage in different lung cell types [162,163], eliciting mitochondrial dysfunction and cell death [164]. Oxidative damage to alveolar epithelial cells causes release of injury molecules, such as damage-associated molecular patterns (DAMPs), through mitochondrial permeability transition pores (mtPTP) that can communicate with the neighboring cells, initiating fibrotic, inflammatory, and immune responses [159,165]. Studies found that hyperoxia-induced mtDNA damage repair mediated by DNA repair enzymes reduced ROS production and apoptosis in alveolar epithelial as well as endothelial cells [164,166,167].

## 8. Mitochondrial Organelle Dynamics in BPD

Mitochondria are highly dynamic organelles. They continuously undergo the processes of fission and fusion (mitochondrial dynamics) to maintain cell homeostasis. Several proteins encoded by the nuclear genome are involved in the process. Inhibition of fusion or activation of fission can lead to mitochondrial fragmentation. On the contrary, activated fusion and/or inhibited fission can trigger mitochondrial hyperfusion/elongation [154]. Mitochondrial fission is mediated by a number of proteins, the most important being dynamin-related protein 1 (Drp1), mitochondrial fission factor, and MiD 49/51 [168], whereas mitochondrial fusion is mediated by mitofusin (MFN) 1 and 2, OMM, and optic atrophy 1 (OPA1, IMM fusion). Oxidative stress causes imbalance in the levels of these proteins, resulting in mitochondrial fragmentation [169]. Pulmonary endothelial cells subjected to hyperoxia exhibited increased phosphorylation of DRP1 (serine 616), OPA1, PINK-1, p62, and LC3B, and decrease in MFN1. The hyperoxia-induced obstruction of pulmonary microvascular development and pulmonary artery pressure was relieved upon treatment with DRP1 inhibitor Mdivi-1 [170]. Hyperoxia is suggested to cause imbalance in the fission/fusion process, causing mitochondrial dysfunction [158]. The changes were mitigated upon returning the cells to normoxia, treatment with antioxidants such as MitoTEMPOL, Drp-1 silencing, or inhibition or protection by the mitochondrial endonuclease ENDO III [171]. Phosphorylation of serine at 637 of Drp1 promotes endoplasmic reticulum–mitochondria interaction, preventing mitochondria from fission [172]. Under stress, SIRT3, a histone deacetylase, deacetylates and activates OPA1, thereby promoting mitochondrial fusion regulating mitochondrial function [173]. Hyperoxia is associated with increased expression levels of PINK1-Parkin and Nip3-like protein X, accompanied by accumulation of dysfunctional mitochondria and mitochondrial autophagy [174].

## 9. Mitochondrial Biogenesis in BPD

Mitochondrial biogenesis is the process by which cells increase their mitochondrial numbers, which ultimately increase mitochondrial metabolic capacity. It is regulated by mitochondria as well as nuclear genes. PGC-1α (peroxisome-proliferator-activated receptor γ co-activator-1α) is a cotranscriptional regulation factor that activates transcription factors including NRF1 and NRF2, which further activate mitochondrial transcription factor A (TFAM) to stimulate mitochondrial biogenesis. TFAM is the key molecule that drives transcription and replication of mitochondrial DNA and is critical for lung development [175]. Mitochondrial biogenesis in the pulmonary vasculature noted following inhalational lung injury was investigated [176]. Clinical drugs like aminophylline and montelukast, used to treat asthma, have the ability to promote mitochondrial biogenesis of alveolar epithelial cells through CREB/PGC-1α[177,178]. These findings suggest that the biogenesis of mitochondria may play a significant role in protecting against hyperoxic lung injury.

## 10. Cellular Respiration in BPD

Cellular respiration, important for the proper functioning of the cells and the survival of organisms, is achieved through ATP generation [179]. The process includes involvement of respiratory chain complexes I–V present on the mitochondrial inner-membrane. When there is inhibition of mitochondrial respiratory chain, impairment in mitochondrial membrane accompanied by uncoupling of oxidative phosphorylation, mitochondrial Ca^2+^ mishandling, or dysregulated mitochondrial ROS generation, failure in mitochondrial bioenergetics occurs [158]. Presence of dysmorphic, swollen mitochondria with inner-membrane abnormalities was noted in in pulmonary cells of animals exposed to chronic hyperoxia [180]. Mitochondria isolated from hyperoxic mice showed reduced complex I and II activities. In endothelial cells, hyperoxia affected complex I activity impairing NADH-dependent mitochondrial respiration [156], affecting pulmonary endothelial cell proliferation, leading to death [181]. In MLE12 cells, which are of a murine lung alveolar epithelial cell line, hyperoxia inhibited basal and maximal respiration [182]. In A549 cells, a type II alveolar epithelium human lung adenocarcinoma cell line exposure to hyperoxia reduced the ATP content and mitochondrial membrane potential [183]. Hyperoxia-induced mitochondrial ROS might modulate ATP production by indirectly reducing the Ca^2+^ buffering capacity and inhibiting mitochondrial respiratory chain, which is attenuated by mitochondria-specific antioxidant, MitoTEMPOL [161]. Mitochondrial ROS independent effects of hyperoxia on cellular respiration are also reported [184]. Inactivation of aconitase and inhibition of the tricarboxylic acid cycle, thereby contributing to hyperoxic lung damage, is reported [185]. Hyperoxia affects mitochondrial lipid synthesis and the metabolism, affecting lung surface tension and compliance [186,187].

## 11. Sphingolipid Signaling in Hyperoxic Mitochondrial Injury

Sphingolipids are important components of cell membrane [188] and play a significant role as signaling molecules in cell growth, differentiation, and apoptosis [115,189,190]. Members of the sphingolipid family including ceramide, sphingosine, S1P and ceramide-1-phosphate (C1P), in addition to their roles as bioactive molecules in various signal transduction pathways [56,57], also adversely modulate mitochondrial function affecting cellular energy and metabolism. Sphingolipids adversely affect various steps involved in redox homeostasis. They activate NOX proteins and induce ROS production, downregulate antioxidant enzymes, and damage mitochondrial integrity [191,192]. Several mitochondrial dysfunction-related conditions are associated with a disturbed sphingolipid metabolism [193,194]. Sphingosine Kinase 1 Inhibitor, PF543, ameliorates mitochondrial DNA damage in lung epithelial cells [79]. In small airway epithelial cells, inactivation of SPHK1 reduced mitochondrial ROS production [29]. Prolonged hyperoxia results in acute lung injury and the shifting balance of apoptosis regulating proteins such as Bcl-2, Bax, and ceramide could contribute to the evolution of the inflammatory process [195]. Increased intracellular concentration of ceramide impedes mitochondrial function by interfering with multiple aspects of mitochondrial respiration, mitochondrial electron transport chain (ETC), oxidative phosphorylation (OXPHOS), and ATP production and mitochondrial biogenesis, including fission–fusion dynamics (Figure 3) [196,197]. Ceramides play a role even in promoting lung cancer cell apoptosis by lethal mitophagy [198]. Ceramides are required for the complex activities of electron transport chains, but they inhibit ETC, inducing ROS generation. Ceramide also initiates mitochondrial outer-membrane permeability and pore formation, reducing the mitochondrial membrane potential. Thus, ceramides are determinants for the induction of mitophagy. Mitophagy is a mechanism by which cell removes dysfunctional mitochondria [199,200]. Ceramide-producing enzymes are reported to be present in mitochondria [201,202,203]. There are abundant reports linking ceramide synthase and ceramides to mitochondrial fission. Mitochondrial fusion is a compensatory mechanism for the cells to survive by mixing the contents of partially damaged mitochondria, decreasing stress, and maintaining ATP generation. Mitochondrial fission is considered as mitochondrial division to generate new mitochondria. Increase in ceramide induces fission proteins, causing mitochondrial fragmentation. Inhibition of ceramide synthase decreases fission-promoting DRP1 and Fis1 recruitment, leading to reduced mitochondrial fission and mitochondrial respiration. These effects are reduced by DRP1 inhibition [204,205,206]. Hyperoxia followed by room air recovery reduced mitochondrial oxidative phosphorylation by decreasing fatty acid oxidation in endothelial cells [207].

## 12. Summary and Conclusions

Bronchopulmonary dysplasia (BPD) and associated complications are one of the major reasons for morbidity and mortality seen in preterm infants [4,7]. Recent studies identified a crucial role for sphingolipid signaling in the pathogenesis of BPD [27,28,29,30,41]. In preterm infants, oxidative stress in poorly developed lungs complicates the disease progression [43]. Oxidative stress induces mitochondrial dysfunction and alveolar developmental arrest in BPD [180]. The pathophysiological events contributing to hyperoxia-induced mitochondrial dysfunction are summarized in Figure 4. Abnormal sphingolipid signaling aggravates BPD pathogenesis by adversely affecting mitochondrial function and redox homeostasis [181]. In addition, sphingosine 1-phosphate (S1P) under hyperoxia tends to promote epithelial-mesenchymal transition (EMT) [208], whereas S1P receptor 1 (S1PR1) mediates pathogenesis of BPD by inhibiting pulmonary vascularization [31]. Inhibition of activated sphingosine kinase 1 (SPHK1)/S1P/S1PR1 signaling axis in animal model of hyperoxia ameliorated the severity of BPD and its sequela of airway remodeling [27,29,30,31]. In this context, we emphasize the importance of SPHK1/S1P/S1PR1 axis as a convincing drug target for the treatment of BPD. Inhibition of enzymes and receptors in this pathway could serve as therapy for lung conditions beyond BPD.

## Figures and Tables

**Figure 1 ijms-23-01254-f001:**
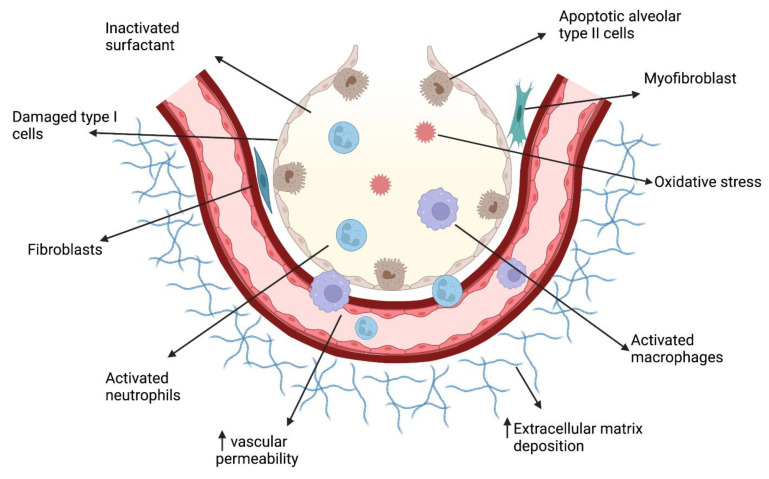
Schematic overview of pathophysiology of bronchopulmonary dysplasia (BPD). BPD, pathogenesis of which is multifactorial, is a culmination of several pathological processes in lung, leading to inflammation, cellular apoptosis, and extracellular matrix remodeling, which affects lung growth, function, immunity, alveolarization, vascularization, and repair and regeneration. The vertical arrows (↑) indicate ‘increase ’.

**Figure 2 ijms-23-01254-f002:**
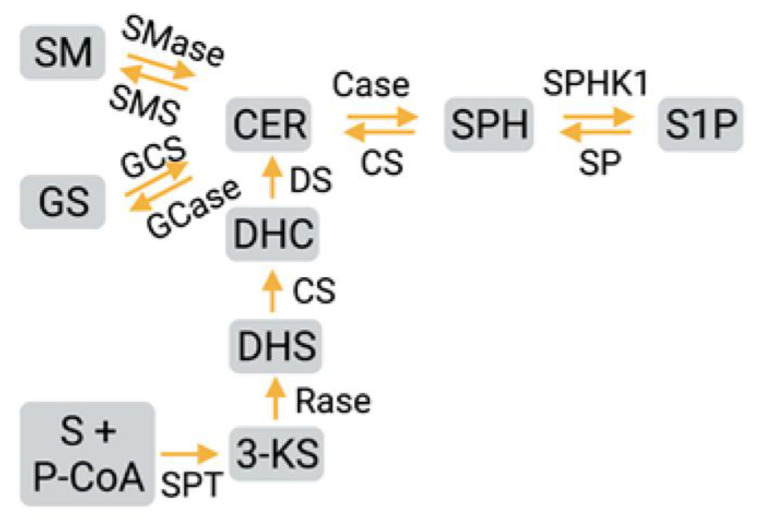
Sphingolipid biosynthesis. Ceramide synthesis occurs through various catalytic reactions from sphingomyelin, glycosphingolipids, as well as serine and palmitoyl coenzyme A, from which sphingosine and sphingosine-1-phosphate are produced; most reactions are reversible. SM: sphingomyelin; GS: glycosphingolipids; CER: ceramide; DHC: dihydroceramide; S: serine; P-CoA: Palmitoyl CoA; SPT: serine palmitoyl transferase; SPHK1: sphingosine kinase 1; CS: ceramide synthase; Case: ceramidase; SP: sphingosine phosphatase; SPH: sphingosine; S1P: sphingo-sine-1-phosphate; SMase: sphingomyelinase; SMS: sphingomyelin synthase; Gcase: glucosylceramidase; GCS: glucosylceramide synthase; 3-KS: 3-ketosphinganine; Rase: reductase; DS: desaturase.

**Figure 3 ijms-23-01254-f003:**
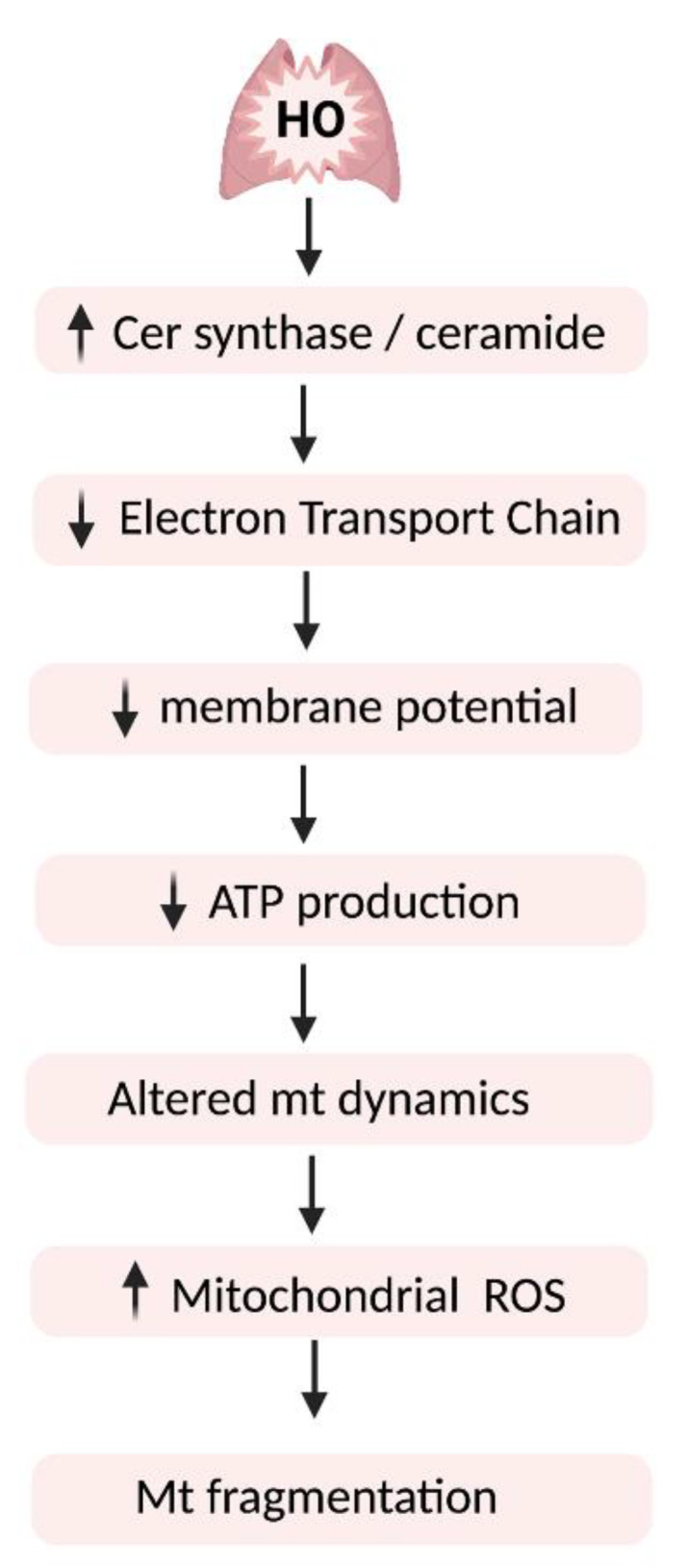
Effects of altered ceramide signaling pathway on mitochondrial dynamics. Hyperoxia (HO)-induced increase in ceramide synthase/ceramide production augments mitochondrial ROS production and impairs cellular respiration, thereby affecting ATP production. This adversely impacts mitochondrial fission/fusion process, leading to mitochondrial fragmentation, and ultimately cell death. The vertical arrows (↑) indicate ‘increase ’ and (↓) indicate ‘decrease ’.

**Figure 4 ijms-23-01254-f004:**
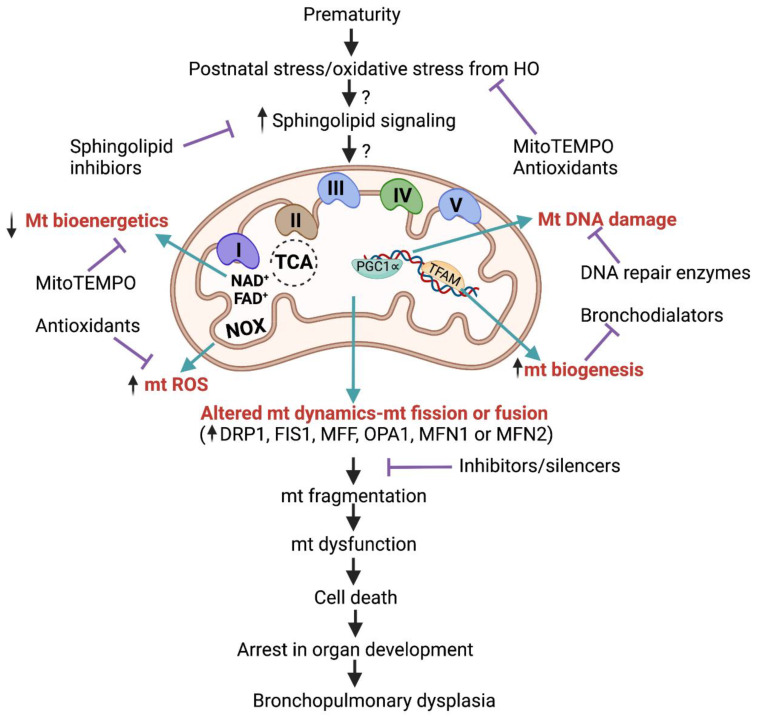
Pathophysiological events contributing to hyperoxia-induced mitochondrial dysfunction. Oxidative stress induced during HO could alter sphingolipid signaling, leading to several events that influence one another, thereby causing mitochondrial dysfunction. Molecules inhibiting each event are potential targets for therapeutic interventions. The vertical arrows (↑) indicate ‘increase ’, (↓) indicate ‘decrease ’ and the 
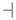
 indicate inhibition.

**Table 1 ijms-23-01254-t001:** Small molecule inhibitors for different lung disease applications.

Specificity	Drugs	Disease Models
S1P1 modulators	Fingolimod (FTY720)	Asthma [93] Lung cancer [94]
	Siponimod	Multiple sclerosis [95]
	NIBR0213	BPD [31]
	SB-649146	ALI [96]
S1P2 modulators	JTE-013	Asthma [97] Pulmonary hypertension [98] Cystic fibrosis [99]
S1P3 modulators	BML-241	Asthma [97]
SK1 modulators	PF543	BPD [29,30] Pulmonaryfibrosis [79,100] Cancer [101]
SK2 modulators	ABC294640 (Opaganib)	Cancer [102]
CERS inhibitors	CERS 2, 4, and 6	Cancer [103]
Cerk inhibitors	NVP-231	Cancer [104]

## Data Availability

Not applicable.

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
