# Peer review of "The Role of Sphingolipid Signaling in Oxidative Lung Injury and Pathogenesis of Bronchopulmonary Dysplasia"

_ijms, 2022, doi:10.3390/ijms23031254_

Round 1

Reviewer 1 Report

1) Abstract: Premature infants are born with developing lungs burdened by surfactant deficiency and

a dearth of antioxidant defense systems. Survival rate of such infants has significantly improved

due to advances in care involving mechanical ventilation and oxygen supplementation. However,

a significant subset of such survivors develop the chronic lung disease, Bronchopulmonary dyspla-

sia (BPD) characterized by enlarged simplified alveoli and deformed airways. Among a host of fac-

tors contributing to the pathogenesis is oxidative damage induced by exposure of the developing

lungs to hyperoxia. Recent data indicate that hyperoxia induces aberrant sphingolipid signaling

leading to mitochondrial dysfunction and abnormal reactive oxygen species (ROS) formation (ROS).

The role of sphingolipids such as ceramides and sphingosine 1-phosphate (S1P), in the development

of BPD has emerged in the last decade. Both ceramide and S1P are elevated in tracheal aspirates of

premature infants of <32 weeks gestational age developing BPD. This was faithfully reflected in the

murine models of hyperoxia and BPD, where there is an increased expression of sphingolipid me-

tabolites both in lung tissue and bronchoalveolar lavage. Treatment of neonatal pups with a sphin-

gosine kinase1 specific inhibitor, PF543 resulted in protection against BPD as neonates accompanied

by improved lung function and reduced airway remodeling as adults. This was accompanied by

reduced mitochondrial ROS formation. S1P receptor1 induced by hyperoxia also aggravates BPD,

revealing another potential druggable target in this pathway for BPD. In this review we aim to pro-

vide a detailed description on the role played by sphingolipid signaling in hyperoxia induced lung

injury and BPD. Please clarify the abstract and divided it in different paragraphs.

2) Introduction L 75-77.  Role of sphingolipid signaling in generation of ROS  and apoptosis is demonstrated [44] and recent studies have highlighted a potential link between sphingolipid signaling inducing ROS formation and BPD [29]. Please improve this paragraph and add these references:

a-  Sarode, P., Mansouri, S., Karger, A., Schaefer, M. B., Grimminger, F., Seeger, W., & Savai, R. (2020). Epithelial cell plasticity defines heterogeneity in lung cancer. Cellular signalling65, 109463. https://doi.org/10.1016/j.cellsig.2019.109463

b- Ruaro, B., Salton, F., Braga, L., Wade, B., Confalonieri, P., Volpe, M. C., Baratella, E., Maiocchi, S., & Confalonieri, M. (2021). The History and Mystery of Alveolar Epithelial Type II Cells: Focus on Their Physiologic and Pathologic Role in Lung. International journal of molecular sciences22(5), 2566. https://doi.org/10.3390/ijms22052566

3) Introduction. L 80-82. In this review, we provide a comprehensive review of the literature giving a better insight in to sphingolipid metabolism and the role of sphingolipid signaling  induced ROS leading to BPD.

Please underline the novelty of your review.

4)Please add a table with the most important features on the topic and the related  references

5) 12. Summary and Conclusion L 427-437. BPD and associated complications are one of the major reasons for morbidity and  mortality seen in preterm infants [4,7]. Recent studies have identified a crucial role for  sphingolipid signaling in the pathogenesis of BPD [27–30,41]. In preterm infants, oxidative stress in poorly developed lungs complicate the disease progression [176]. Oxidative  stress induces mitochondrial dysfunction and alveolar developmental arrest in BPD [149]. Abnormal sphingolipid signaling aggravates BPD pathogenesis by adversely effecting mitochondrial function and redox homeostasis [150]. In addition, S1P under hyperoxia tends to promote epithelial- mesenchymal transition (EMT) [177] whereas S1PR1 mediates pathogenesis of BPD by inhibiting pulmonary vascularization [31]. Ameliorating  SPHK1/S1P/S1PR1 signaling axis in animal model of hyperoxia reduced BPD pathogenesis [27,29–31]. In this context we emphasise the importance of SPHK1/S1P/S1PR1 axis as a 438 convincing drug target for the treatment of BPD. Please underline the clinical applications of your study.

Author Response

We are extremely grateful to the reviewer for the insightful comments. Attention has been paid to all the comments and changes made as suggested. One suggestion was to change the pattern of abstract and split the same into paragraphs which goes against the pattern in IJMS. But for that we have implemented all the suggested changes.

  1. The abstract has been kept unmodified. We noted that none of the published manuscripts in IJMS have the abstract classified into paragraphs. This pattern may be against the pattern preferred by the journal. If the journal editor prefers, we can change the way the abstract is written and change the same into multiple paragraphs.
  2. Introduction L 75-77. The sentences improved and new references added.
  3. Introduction L 80-82. Sentence reworded stressing the importance of the novelty.
  4. Table added with salient features of the manuscript as suggested by another reviewer too.
  5. Summary and Conclusion L 427-437. Reworded as suggested.

Reviewer 2 Report

In the article ijms-1547221, the authors made substantial research efforts to survey the literature and summarize the latest updates on the role of sphingolipid signaling in hyperoxia induced lung injury and  Bronchopulmonary dysplasia. Moreover, a detailed classification and explanation of the different approaches, and mechanisms of ROS formation and antioxidant defense have been well described. The authors emphasized and illustrated the mitochondrial dynamics, biogenesis and oxidative stress and their roles in BPD and mitochondrial injury. In addition, the authors showed and described the role of sphingolipid signaling in hyperoxic mitochondrial injury and  BPD pathogenesis.

Overall, the surveyed literature and protocol applied to achieve its purpose are adequate. The information provided by the manuscript could be a gain for researchers interested in understanding the role of sphingolipids in lung diseases. Accordingly, I would recommend the publication of this article after the following revision and suggestions:

- the introduction about sphingolipids is poorly presented. I suggest that the authors add a short part of introduction about it in the beginning of the article and include a short fig for metabolism. This would make it more interesting for readers.

- One of the major concern in this article, the main message of this article is the role of sphingolipids in lung diseases and BPD, however, this has been only surveyed in section 4 and 11?? and the no of surveyed studies is not relevant, almost 40-50 studies (out of 177) are cited in this article which are related to this topic. The major covered studies are mainly relevant to mitochondrial injure,ROS,... etc. Accordingly, I recommend that the authors better cover the role of sphingolipids in more details and include more of relevant studies..e.g role of sphingomyelin, sphingomyelinases, more recent studies about S1P.

- The authors should also include a section which describe in details the role of targeting sphingolipid metabolism enzymes and the application of different SPL inhibitors in lung diseases and BPD. Including tables is recommended..

- the authors should also give a short intro for the prospective role of deoxysphingolipids in lung diseases. This class of sphingolipid has been recently shown to play an essential role in mitochondrial balance and activity.

Author Response

We are extremely grateful to the reviewer for the highly insightful comments. We have paid attention to each of the comments and changes have been made as suggested.

  1. The introduction about sphingolipids is poorly presented. I suggest that the authors add a short part of introduction about it in the beginning of the article and include a short fig for metabolism. This would make it more interesting for readers.

Response: We agree with the comment. The introduction about sphingolipids has been added in the introduction. A figure explaining the metabolism also has been added.

  1. One of the major concerns in this article, the main message of this article is the role of sphingolipids in lung diseases and BPD, however, this has been only surveyed in section 4 and 11?? and the number of surveyed studies is not relevant, almost 40-50 studies (out of 177) are cited in this article which are related to this topic. The major covered studies are mainly relevant to mitochondrial injury, ROS, etc. Accordingly, I recommend that the authors better cover the role of sphingolipids in more detail and include more of relevant studies. e.g role of sphingomyelin, sphingomyelinases, more recent studies about S1P.

Response: This is a very relevant suggestion. In section 3, we have given an outline of the role of sphingolipids in diseases in general. We have expanded this section further by adding an extra paragraph. The main stress of the article is on BPD set in the context of other lung diseases so that a future direction could be defined. Not much is known in BPD and the developing lungs.

  1. The authors should also include a section which describe in detail the role of targeting sphingolipid metabolism enzymes and the application of different SPL inhibitors in lung diseases and BPD. Including tables is recommended

Response: We have followed this suggestion and have included a table detailing the therapeutic role in various conditions.

  1. The authors should also give a short intro for the prospective role of deoxysphingolipids in lung diseases. This class of sphingolipid has been recently shown to play an essential role in mitochondrial balance and activity.

Response: This is a wonderful suggestion and has been duly followed. A separate paragraph has been added under section 3. A significant role has been noted for deoxysphingolipids mainly in neurological disorders.

Reviewer 3 Report

 The manuscript by Thomas et al. aims to provide a detailed description on the role played by sphingolipid signaling in hyperoxia-induced lung  injury and,  particularly, in bronchopulmonary dysplasia. The topic is interesting and  the manuscript is clearly written. However, there are some mistakes that should be corrected and some sentences that might be rewritten before the manuscript could be acceptable:

Reference [ 85] is missing in the text.

The manuscript includes 3 figures, but there is no information of the placement of the figures in the text.

Figures 2,3: Mitochondrial is abbreviated either as mt or as Mt. Only one abbreviation should be used, and the meaning of the abbreviation mt/Mt should be added to the legend.

153: The Authors claim that “FTY720 remains as a promising drug in the  treatment of Type II diabetes mellitus”. The reference cited is not recent. There is any clinical trial on FTY720 and type II diabetes mellitus?

Unnecessary abbreviations should be eliminated. 184: airway remodeling (AWRM); 279: necrotizing enterocolitis (NEC); 426 fatty acid oxidation (FAO) in endothelial cells.

205 “Based on these results we hypothesize that S1PR1 modulator, FTY720, could be a promising drug in the treatment of BPD”

Since S1P and FTY720 have cardiovascular effects (https://doi.org/10.1016/j.ahj.2014.06.028),  I suggest to the Authors to add a comment on putative side effects of FTY720 on premature infants.

308 “Mitochondrial dysfunction has recently been recognized to play a significant role in  the pathogenesis of various diseases”. Reference is missing.

333-336 Could the Authors specify the roles of SIRT3 and PINK on mitochondrial and cell functions? Did hyperoxia increase or decrease mitophagy? Did Opa deacetylation increase or decrease fusion?

86 is not adequate for sphingosine kinase 2. Please check. Sphingosine kinase 2 was cloned ans characterized by  Liu et al. doi:10.1074/jbc.m002759200

366- 367 References 146 and 147 (doi:10.3390/molecules19044967) do not seem adequate for this statement: “Clinical drugs like aminophylline and  montelukast, used to treat asthma have the ability to promote mitochondrial biogenesis of alveolar epithelial cells through CREB/PGC-1α” [146,147]. I guess that Authors refer to other papers by Wang et al. and Wei et al.  doi: 10.1016/j.bbrc.2019.05.013 and doi: 10.1080/21691401.2019.1687502

383 Information regarding the cell lines should be provided. It is interesting to the reader to know that MLE12 is a murine epithelial cell line and A549 is a type II alveolar epithelium human lung adenocarcinoma cell line

400 They activate NOX proteins and induce ROS production, down regulate antioxidant enzymes, and destructs mitochondrial integrity. Please correct “destructs”

401-2: Several mitochondrial dysfunction-related conditions are associated with a disturbed sphingolipid  metabolism [163,164]

The two references indicated do not seem adequate for “mitochondrial dysfunction-related conditions. The Authors could add an statement regarding the role of ceramide on lung cancer, or they should replace  these references with  more suitable references, such as doi: 10.1038/cdd.2017.128 and [166]

411-412 Ceramides are required for the complex activities of electron transport chain but inhibits ETC inducing  ROS generation. Please correct “inhibits”

436-437 “Ameliorating  SPHK1/S1P/S1PR1 signaling axis in animal model of hyperoxia reduced BPD pathogenesis [27,29–31]”.

The sentence is not clear. What do the Authors mean with ameliorating? The Authors have described that knockout of Sphk1 is protective. The Authors should clearly indicate that it is the reduction of the activity which leads to protection.

Author Response

We are extremely grateful to the reviewer for the highly insightful comments. We have paid attention each of the comments and changes have been made as suggested.

  1. Reference [ 85] is missing in the text.

Response: This has been rectified. As additional references are added as suggested by reviewers, this has become reference [96] now.

  1. The manuscript includes 3 figures, but there is no information of the placement of the figures in the text.

Response: Figures have been appropriately mentioned in the text.

  1. Figures 2,3: Mitochondrial is abbreviated either as mt or as Mt. Only one abbreviation should be used, and the meaning of the abbreviation mt/Mt should be added to the legend.

Response: This error has been rectified.

  1. 153: The Authors claim that “FTY720 remains as a promising drug in the treatment of Type II diabetes mellitus”. The reference cited is not recent. There is any clinical trial on FTY720 and type II diabetes mellitus?
  2. Response: There are only animal studies showing protection by FTY 720 against diabetes mellitus and no clinical trials have been done. there are no recent studies either other than what we have mentioned. We have added an extra sentence clarifying that no clinical trials have been reported.

  1. Unnecessary abbreviations should be eliminated. 184: airway remodeling (AWRM); 279: necrotizing enterocolitis (NEC); 426 fatty acid oxidation (FAO) in endothelial cells.

Response: This error has been rectified. All abbreviations have been removed.

  1. 205 “Based on these results we hypothesize that S1PR1 modulator, FTY720, could be a promising drug in the treatment of BPD”. Since S1P and FTY720 have cardiovascular effects https://doi.org/10.1016/j.ahj.2014.06.028), I suggest to the Authors to add a comment on putative side effects of FTY720 on premature infants.

Response: This suggestion has been implemented by adding clarification and comment.

  1. 308 “Mitochondrial dysfunction has recently been recognized to play a significant role in the pathogenesis of various diseases”. Reference is missing.

Response: References have been added as suggested.

  1. 333-336 Could the Authors specify the roles of SIRT3 and PINK on mitochondrial and cell functions? Did hyperoxia increase or decrease mitophagy? Did OPA deacetylation increase or decrease fusion?

Response: SIRT3 deacetylates and activates OPA1 and mitochondrial fusion. Hyperoxia ultimately leads to increased mitochondrial autophagy. The sentences are reworded to reflect this accurately.

  1. 86 is not adequate for sphingosine kinase 2. Please check. Sphingosine kinase 2 was cloned as characterized by Liu et al. doi:10.1074/jbc.m002759200

Response: This error has been rectified and new reference added.

  1. 366- 367 References 146 and 147 (doi:10.3390/molecules19044967) do not seem adequate for this statement: “Clinical drugs like aminophylline and montelukast, used to treat asthma have the ability to promote mitochondrial biogenesis of alveolar epithelial cells through CREB/PGC-1α” [146,147]. I guess that Authors refer to other papers by Wang et al. and Wei et al. doi: 10.1016/j.bbrc.2019.05.013 and doi: 10.1080/21691401.2019.1687502

Response: This error has been rectified and new references added.

  1. 383 Information regarding the cell lines should be provided. It is interesting to the reader to know that MLE12 is a murine epithelial cell line and A549 is a type II alveolar epithelium human lung adenocarcinoma cell line

Response: This has been explained as suggested.

  1. 400 They activate NOX proteins and induce ROS production, down regulate antioxidant enzymes, and destructs mitochondrial integrity. Please correct “destructs”

Response: The word destructs has been replaced by damage to read as “damages mitochondrial integrity”.

  1. 401-2: Several mitochondrial dysfunction-related conditions are associated with a disturbed sphingolipid metabolism [163,164]. The two references indicated do not seem adequate for “mitochondrial dysfunction-related conditions. The Authors could add an statement regarding the role of ceramide on lung cancer, or they should replace these references with more suitable references, such as doi: 10.1038/cdd.2017.128 and [166]

Response: This suggestion has been implemented. The suggested reference added and an extra sentence on ceramide added.

  1. 411-412 Ceramides are required for the complex activities of electron transport chain but inhibits ETC inducing ROS generation. Please correct “inhibits”

Response: This error has been rectified.

  1. 436-437 “Ameliorating SPHK1/S1P/S1PR1 signaling axis in animal model of hyperoxia reduced BPD pathogenesis [27,29–31]”. The sentence is not clear. What do the Authors mean with ameliorating? The Authors have described that knockout of Sphk1 is protective. The Authors should clearly indicate that it is the reduction of the activity which leads to protection.

Response: This error has been rectified. The sentence has been reworded.

Round 2

Reviewer 2 Report

Thanks to the authors for addressing major of concerns that have raised in the first report. However, there are minor concerns that still need to be revised.

1- regarding the introduction about sphingolipids; in the added paragraph, reference 53 is not relevant to the paragraph/stated information. Also, in the new figure added (Fig2), please correct it. From S+P-COA to DHC there are 3 enzymatic reactions not only 2, SPT, 3-KDHS, and CerS. I would be also suggested to cite some recent reviews that cover the most recent findings about sphingolipid metabolism. e.g., 10.3390/ijms22115793, 10.1038/nrm.2017.107

2- the application of different SPL inhibitors in lung diseases and BPD is not fully covered. Also, table 1 does not fully summarize all inhibitors that have been applied. Further, please correct the table. Under the drug column should be the inhibitors, not enzyme/target protein/mode of action. 

3- The authors have adequately introduced the deoxySL to their topic. However, it would be suggested that the authors cite some recent studies in their new paragraph e.g., 10.1016/j.devcel.2021.10.018, 10.3390/ijms22158171, , 10.1194/jlr.M067033, 10.1194/jlr.M072421.

Author Response

Thank you for the valuable suggestions which had improved the quality of our manuscript significantly.

1- regarding the introduction about sphingolipids; in the added paragraph, reference 53 is not relevant to the paragraph/stated information. Also, in the new figure added (Fig2), please correct it. From S+P-COA to DHC there are 3 enzymatic reactions not only 2, SPT, 3-KDHS, and CerS. I would be also suggested to cite some recent reviews that cover the most recent findings about sphingolipid metabolism. e.g., 10.3390/ijms22115793, 10.1038/nrm.2017.107

Response:  Reference 53 as per revision 1 version is a review article from our group on the “Advancements in Understanding the Role of Lysophospholipids and Their Receptors in Lung Disorders Including Bronchopulmonary Dysplasia” and is one of the papers we referred to for the sphingolipid metabolism. We have also added the additional references suggested by the reviewer.

Thank you for the suggestion. Figure 2 has been modified as suggested by including all 3 enzymatic reactions.

Thank you for your valuable input. We have included the recent findings about sphingolipid metabolism as suggested by the reviewer.

2- the application of different SPL inhibitors in lung diseases and BPD is not fully covered. Also, table 1 does not fully summarize all inhibitors that have been applied. Further, please correct the table. Under the drug column should be the inhibitors, not enzyme/target protein/mode of action. 

Response: Thank you for the suggestion and we have modified both the table as advised.

3- The authors have adequately introduced the deoxySL to their topic. However, it would be suggested that the authors cite some recent studies in their new paragraph e.g., 10.1016/j.devcel.2021.10.018, 10.3390/ijms22158171, 0.1073/pnas.2002391117, 10.1194/jlr.M067033, 10.1194/jlr.M072421

Response: The additional references suggested by the reviewer has been added in the new edited version of manuscript.